# Exploring the Bacterial Community in Aged Fecal Sources from Dairy Cows: Impacts on Fecal Source Tracking

**DOI:** 10.3390/microorganisms11051161

**Published:** 2023-04-28

**Authors:** Megan L. Devane, William Taylor, Pierre-Yves Dupont, Bridget Armstrong, Louise Weaver, Brent J. Gilpin

**Affiliations:** Health and Environment Group, Institute of Environmental Science and Research, Christchurch 8041, New Zealandbridget.armstrong@esr.cri.nz (B.A.); louise.weaver@esr.cri.nz (L.W.); brent.gilpin@esr.cri.nz (B.J.G.)

**Keywords:** fecal source tracking, cowpat, metagenomics, fecal bacterial decay, 16S rRNA, FEAST

## Abstract

(1) Background: This paper discusses the impact of agricultural activities on stream health, particularly in relation to dairy cow fecal pollution. The study explores the fecal microbiome of cattle and the potential ecological implications of aging fecal pollution on waterways. (2) Methods: The study examines changes in the bacterial community available for mobilization from in-situ decomposing cowpats and the effects of simulated rainfall. The microbiome of individual cowpats was monitored over 5.5 months. We used 16S rRNA metagenomics and machine learning software, FEAST (Fast Expectation-mAximization for microbial Source Tracking), for bacterial and fecal source assignments. (3) Results: The phyla Bacillota and Bacteroidota are dominant in the fecal microbiota of fresh cow feces but shift to Pseudomonodota, Actinomycetota, and environmental Bacteroidota in aged cowpats. Potential impacts of these bacterial community shifts on inputs to local agricultural streams are discussed in relation to water quality monitoring and aging sources of fecal contamination. We identified taxon orders that are potential indicators of fresh cattle sources (Oscillospirales and Bacteroidales) and aged sources (Peptostreptococcales-Tissierellales) in water bodies. (4) The paper highlights that bacterial metagenomic profiling can inform our understanding of the ecology of microbial communities in aquatic environments and the potential impacts of agricultural activities on ecosystem health.

## 1. Introduction

The degradation of streams impacted by agricultural activities has caused worldwide concern, particularly with increases in intensive farming. Livestock fecal pollution can be a substantial source of direct and diffuse contamination from overland runoff and subsurface tile drainage into waterways, contributing to elevated concentrations of fecal indicator bacteria such as *Escherichia coli* and potentially pathogens [1,2,3].

The fecal microbiome of cattle has been explored to characterize its diversity and variability between farming operations and feed [4,5,6]. Investigations of the fecal microbiome can provide potential insights into the ecological implications of fecal pollution impacts on waterways [4,7]. With the advent of next-generation sequencing techniques, researchers have postulated that microbial community analyses could provide insights into aquatic ecosystems and even be predictors of stream ecological health [8,9].

New metagenomic software that uses machine learning, such as SourceTracker and FEAST, is being trialed as a useful approach for fecal source assignments using the microbial community identified in water samples (the sink) and comparing it with the microbial community profiles identified in animal fecal sources (the sources) [10,11,12,13]. A recent study using SourceTracker noted that while this program has value in community profiling microbial source tracking studies, there is still a lot of research required to understand its limitations, and they advised using local community fecal libraries relevant to the known fecal sources [14]. It was also noted that predictions were improved where source profiles had lower variability of taxa within a specific animal fecal source.

A study in urban and agricultural aquatic environments investigated whether anthropogenic activities resulting in stream degradation could also impact in-stream microbial communities [8]. Water quality parameters, including chemical and biological indexes, were assessed in conjunction with 16S rRNA amplicon sequencing of stream microbial communities to reveal land use impacts. The authors proposed that microbial community changes could be used to assess ecosystem health in response to man-made impacts on the surrounding catchment. In the study, certain taxa, such as Burkholderiales and Verrucomicrobia, were identified as more abundant and prevalent in streams in poor condition.

Shifts in the bacterial community of decomposing cowpats have been observed over time [7,15]. Therefore, during weather events, aged fecal sources prone to land runoff into streams should be factored into the impacts on the aquatic bacterial community and the introduction of pathogens relevant to health risk assessments [16]. A eukaryotic microbial community assessment of flooding impacts on aquatic environments noted a marked and sustained change in the microbial community post-flood [17]. An understanding of the natural ecology of microbial communities in aquatic environments and recognition of microbial inputs from fecal pollution are important steps in developing tools to assess ecological stream health impacts on the abundance of taxa associated with anthropogenic activities. These assessments will become more pressing as the impacts of climate change intensify with heavy rain and flooding events.

A previous study of the influence of flood and rainfall on the mobilization of bacteria from cowpats noted marked reductions in the concentration of fecal source markers (*E. coli*; microbial source tracking markers: GenBac3, BacR, and CowM2; and fecal sterol markers) as the cowpats decomposed under summertime field conditions over more than five months [18]. This current study investigates the bacterial community composition in DNA extracts from the previous cowpat study [18] using 16S rRNA metagenomics and the machine learning software FEAST (v0.1.0) (Fast Expectation-mAximization for microbial Source Tracking) for fecal source assignments. Changes in the bacterial community mobilized from aging cowpats are compared to the bacterial community mobilized during simulated rainfall. The potential impacts of these shifts in bacterial communities on inputs to local agricultural streams are discussed in relation to water quality monitoring.

## 2. Materials and Methods

### 2.1. Sampling of Cowpats

This study describes the metagenomic analysis of DNA extracted from decomposing cowpats as previously described in the paper of Devane et al. [18]. In brief, fresh fecal material from 60 pasture-fed dairy cows was collected from the concrete pad leading into the milking shed, which had been washed down prior to milking. The fecal slurry was homogenized prior to the preparation of 70 simulated 1 kg cowpats. These simulated cowpats were prepared by pouring fecal slurry into a sterile plastic ring sitting on sterilized nylon mesh and were placed on a mix of ryegrass and clover within a secure enclosure in Christchurch, NZ. This outdoor setting had not previously been grazed by livestock and was free of fecal material. On each of the sampling days 1, 8, 15, 22, 29, 50, 71, 105, 134, and 162 (post-deposition), three individual, entire cowpats were sampled for analysis of the cowpat microbial reservoir, three cowpats for rainfall runoff simulation, and a seventh cowpat was used to determine dry weights for moisture loss analysis. For this time series experiment, therefore, there were three cowpat samples and 10 sampling intervals per treatment (cowpat and rainfall runoff), which equates to 30 samples per treatment group.

### 2.2. FEAST eDNA Source Library

Ten samples of grass (ryegrass and clover) and underlying soil (top 2 cm) were collected for bacterial community analysis for the FEAST source library. These environmental samples were collected from the same secure enclosure where the cowpat experiment was undertaken. At the perimeter of the secure enclosure, a 5 cm diameter brass borer was used to take 10 soil cores to a depth of approximately 2 cm. The cores were taken and included all the grass and herbage on the surface. The soil borer was sterilized with alcohol wipes between cores, and the cores were transported to the laboratory in sterile plastic containers. Processing began within one hour of sampling. Fecal samples (10–50 g) from freshly deposited cowpats were collected from dairy cows nationwide on New Zealand farms and transported at 4–8 °C. Care was taken not to touch the soil or grass during collection, and fecal samples were processed within 24 h of collection. To align with the need for similar-sized source libraries for FEAST analysis, ten of these fecal samples were chosen for inclusion as the cow fecal source.

### 2.3. Total Microbial Reservoir in the Cowpat

To release the microbial reservoir from entire cowpats, at each sampling interval, three cowpats were individually suspended in sterile distilled water (MilliQ, Merck Millipore, Darmstadt, Germany) to a final weight of 2 kg. Stirring the cowpat with a sterile broom handle for 10 min mobilized microbes from the cowpat into suspension. Encrustation of the cowpats in later stages required manual breaking up of the dried cowpat to maximize the release of microbes and fecal markers. After a settling period of 10 min, supernatants were collected for DNA extraction.

### 2.4. Simulated Rainfall Analyses

A rainfall simulator was constructed [18], which delivered a rainfall event of 20 mm/h with the formation of <2 mm raindrops at terminal velocity [19]. In brief, sterile distilled water (1146 mL) (MilliQ, Merck Millipore, Darmstadt, Germany) was added to a circular drip tray containing 25 needles of 20-gauge size (nominal internal diameter of 0.603 mm). The drip tray with needles was placed 92 cm above the cowpat samples, and wind disturbance during simulated rain was minimized by wrapping the rainfall simulator in plastic [18]. On each sampling occasion, three cowpats were individually collected from the field by lifting up the nybolt mesh plus cowpat and transferring it to a pre-weighed 450 × 300 mm tray. This tray with the cowpat had an approximate 10% slope to facilitate collection, with four 10 mm holes and nine 3 mm holes drilled at one end to allow the rainfall runoff to flow through a sterile funnel into sterile 500 mL polypropylene collection bottles. An autoclaved nybolt mesh in the funnel prevented the collection of insects, grass, leaves, etc. The bottles were placed into holes in the ground to directly capture the rainfall runoff. The measured volume of collected runoff was recorded, and DNA extraction was performed as outlined in the following section.

### 2.5. DNA Extraction of Cowpats and Rainfall Runoff

DNA was extracted using the ZR Fecal DNA Kit™ (Zymo Research, Orange, CA, USA), with initial processing of feces from the early stages of cowpat decomposition and/or filter(s) in a bead beater (MixMate, EppendorfAG, Hamburg, Germany) for 5 min at 2000 g, with further details provided in Devane et al. [18]. In the early stages of cowpat decomposition, when cowpats were less dried out, the supernatant from the cowpat runoff and the collected rainfall runoff (0.3–2 mL) were centrifuged at 4500 g for 10 min. The fecal residue (150 mg) was weighed into bead beater tubes with 750 µL of lysis buffer containing β-mercaptoethanol (Sigma-Aldrich Co., St. Louis, MO, USA). In the latter stages, there was less fecal material re-suspended from the dried-out cowpats, and therefore, 50–600 mL was filtered through 47 mm, 0.45 μm cellulose ester membrane filters (MilliQ, Merck Millipore, Darmstadt, Germany) and re-suspended in bead beater tubes from the Zymo Fecal DNA Kit™ (Zymo Research, Orange, CA, USA) following kit instructions. Blanks for cowpat runoff were 150 mg of UltraPure™ DNase/Rnase-free distilled water (Thermo Fisher Scientific, Waltham, MA, USA) added to bead beater tubes from the Zymo Fecal DNA Kit™ (Zymo Research, Orange, CA, USA). For the rainfall runoff blank, approximately 1100 mL of sterile MilliQ water (Merck Millipore, Darmstadt, Germany) was run through the rainfall simulator prior to each sampling occasion. Rainfall runoff blanks were filtered through 47 mm, 0.45 μm cellulose ester membrane filters (Merck Millipore, Darmstadt, Germany), and DNA extraction of this rainfall control followed the same protocol as the rainfall runoff fecal samples.

### 2.6. DNA Extraction of Soil, Grass, and Fecal Samples for DNA Library and FEAST Analysis

In the laboratory, the grass was aseptically removed from the soil, weighed (range: 1.2 to 2.5 g wet weight), and transferred to a sterile bottle. 100 mL of sterile MilliQ water (Merck Millipore, Darmstadt, Germany) was added to the grass and shaken by hand for 5 min to resuspend bacteria on the surface of the grass. To remove soil particles adhered to grass, a Whatman GF/C filter (Whatman, GE Healthcare Services, Buckinghamshire, UK) was placed on top of the Supor 0.22 µm Polyethersulfone (PES) filter (Pall Corp., Washington Port, NY, USA), and 50 mL of the grass resuspension was filtered with replacement of the Whatman filter when it clogged. The supor filter was extracted for DNA using the DNeasy PowerSoilPro kit (QIAGEN, Venlo, The Netherlands) with the bead beating protocol as outlined by the manufacturers using a MixMate bead beater (MixMate, Eppendorf, Hamburg, Germany). Each soil core was individually broken up and homogenized with a spatula. A sample was taken for dry weight, and approximately 250 mg of soil was extracted using the bead beating protocol of the PowerSoilPro kit (QIAGEN, Venlo, The Netherlands). The soil blank was 250 mg of UltraPure™ Dnase/Rnase-free distilled water (ThermoFisher Scientific, Waltham, Massachusetts, USA) added to a PowerSoilPro (QIAGEN, Venlo, The Netherlands) bead beating tube with 800 µL of CD1 kit lysis buffer. The water blank was 100 mL of sterile MilliQ water (Merck Millipore, Darmstadt, Germany) put through a 0.22 µm Supor Polyethersulfone (PES) filter (Pall Corp., Washington Port, NY, USA) and added to a bead beating tube and treated the same as the soil blank. Fecal samples from animals for the DNA fecal library and FEAST analysis were collected, and approximately 250 mg of feces was extracted using the bead beating protocol of the DNeasy PowerSoilPro kit (QIAGEN, Venlo, The Netherlands) as outlined for the soil DNA extraction.

### 2.7. Bacterial Metagenomic Analysis of 16S rRNA

The bacterial community analysis of cowpats, grass, soil, and fecal samples from other animals targeted the 16S rRNA primers V3–V4 [20] using standard Illumina protocols for preparation and targeting the 16S rRNA amplicon with Forward Primer with Illumina adapter, 5′TCGTCGGCAGCGTCAGATGTGTATAAGAGACAG-CCTACGGGNGGCWGCAG; and 16S Amplicon PCR Reverse Primer with Illumina adapter, 5′GTCTCGTGGGCTCGGAGATGTGTATAAGAGACAG-GACTACHVGGGTATCTAATCC [21]. Illumina MiSeq 2 × 150 bp sequencing was conducted at NZ Genomics Ltd. (Hamilton, New Zealand).

### 2.8. Bioinformatic Processing of Sequencing Data

Amplicon data were analyzed using the R-based Divisive Amplicon Denoising Algorithm-2 (DADA2) v1.18 software package [21,22]. The median number of reads was 62,412 per sample. Sequencing reads were quality checked with fastqc v0.11.7 [23] and multiqc v1.10.1 [24] and quality controlled with bbduk v38.90 [25], removing all unpaired sequence reads and reads containing ambiguous bases or less than 75 bp in length after adapter removal and trimming regions < Q20. Taxonomy assignment was performed using the SILVA database v138.1 [26], sequences belonging to chloroplast or mitochondrial DNA were removed, and further analyses were performed using the phyloseq v1.22.3 R package [27]. It is noted that taxonomic name changes at the phylum level are occurring, and new phyla names according to Oren and Garrity [28] are used, with synonyms to these new names provided when first encountered in the Results and Discussion sections. FEAST (v0.1.0) was used to estimate the fraction of the sink community that was contributed by each of the supplied source environments and to calculate the potential fraction attributed to unknown sources [13].

### 2.9. Statistical Analyses

Statistical analyses were carried out in R v4.0.5 and using the R package vegan v2.6-2 [29]. A Shapiro-Wilks test was performed to test for normality, and a Student’s *t*-test was used to test for significance between the Shannon diversity of samples with the R stats package. SIMPER analysis using vegan v2.6-2 identified the taxa contributing to dissimilarities between sampling days to test the significance of differences between sampling intervals over time [29].

## 3. Results

### 3.1. Bacterial Community Analysis Using Amplicon Metagenomics

This study compared the amplicons of the 16S rRNA from cowpats and rainfall runoff for each of the ten sampling intervals over a period of 5.5 months while cowpats were decomposing under field conditions. The same four phyla were identified as the dominant phyla in both the cowpat and rainfall runoff matrices: Pseudomonodota (synonym Proteobacteria), Bacillota (synonym Firmicutes), Bacteroidota (synonym Bacteroidetes), and Actinomycetota (synonym Actinobacteria) (Figure 1, Appendix A) [28]. In rainfall runoff, the relative abundance (RA) of Pseudomonodota (56%) was much higher than Bacillota (17%); however, these phyla were proportionally similar in cowpats (30% and 28%, respectively). In total, 26 phyla were identified in each matrix; however, the bottom 21 phyla represented a combined total of only 3% and 2% of the total relative abundance in cowpat and rainfall runoff, respectively.

#### 3.1.1. Phyla in Cowpat and Rainfall Runoff

Day 1 bacterial community analysis of the two matrices was similar, with an average RA of the two main phyla in fresh cowpats: Bacillota 73% and 78% and Bacteroidota 21% and 17% for the cowpat and rainfall runoff, respectively (Figure 1, Appendix A). From the next sampling interval, Day 8, onwards, there was a gradual decrease in the abundance of Bacillota in the cowpat (Day 8, 75% RA), with a final RA on Day 162 of 12%. In comparison, in the rainfall runoff, there was a sharp decrease in Bacillota on Day 8 to 6%, with further fluctuations between 2% and 10% for the remainder of the experiment.

In contrast to Bacillota, the RA of Bacteroidota fluctuated throughout the experiment from Day 8 onwards, with a minimum of 5% in both matrices and a maximum of 32% to 33% on Day 71 in cowpat and rainfall, respectively. The lowest RA of Bacteriodota in rainfall runoff occurred on Day 8, similar to the substantial decrease noted for Bacillota in rainfall on the same day. Pseudomonodota, however, showed marked increases in cowpat and rainfall runoff from approximately 2% RA on Day 1 to 11% and 87% on Day 8, respectively. Pseudomonodota in the cowpat increased to 45% on Day 22 and thereafter stayed above 26% RA. The marked increase (87%) in Pseudomonodota in the rainfall runoff on Day 8 was followed by a sustained decrease with fluctuations between 45% and 60% and a RA of 44% by the end of the experiment.

The RA of Actinomycetota also increased from ~1% in both matrices on Day 1. In the cowpat, Actinomycetota increased gradually to a maximum of 33% RA by Day 105 and 28% on Day 162. In the rainfall runoff, this phylum reached a maximum of 24% RA on Day 29, thereafter declining to 17% by the end of the experiment. Verrucomicrobiota was another phylum noted above 1% RA on Day 1 in the cowpat matrix, fluctuating between 0.05% and 1.8% until the end of the experiment. In contrast, Verrucomicrobiota in the rainfall runoff was 0.3% RA on Day 1 and increased to 3% on Day 15 and 7% on the last day of sampling.

#### 3.1.2. Bacterial Taxa at the Class Level

Similarities in class taxa were observed for both matrices on Day 1; however, over time, taxa differentiated both within and between matrices (Figure 2, Appendix A). On Day 1 in the cowpat and the rainfall runoff, respectively, the majority of the phylum Bacillota was represented by the classes Clostridia (67% and 72% RA) and Bacilli (both 5% RA), whereas the phylum Bacteroidota was represented by the class Bacteroidia (21% and 17% RA, respectively). In the cowpat, the Bacilli ranged between 5–17% RA up to Day 50 and then decreased, whereas in the rainfall runoff, the RA of Bacilli reached its peak of 7% on Day 29.

In cowpats, Clostridia decreased from its peak on Day 1 (67% RA) at a steady rate till Day 71 (4% RA) and then fluctuated from a maximum of 19% down to 9% on Day 162. In contrast, in the rainfall runoff, Clostridia decreased to 1% on Day 8 and fluctuated from 0.3% to 4% throughout the remainder of the experiment. The second most abundant class in fresh cowpats, Bacteroidia, fluctuated throughout the experiment, with peaks of 32% and 33% in both matrices on Day 71, when the other dominant class in fresh feces, Clostridia, was detected at its lowest abundance. By the end of the experiment, Bacteroidia were at 27% and 20% RA in cowpat and rainfall runoff, respectively.

From low RA on Day 1 (<1.5%), Gammaproteobacteria, Alphaproteobacteria, and Actinobacteria were detected in increasing RA in both matrices as the experiment progressed. Gammaproteobacteria had the highest RA on Day 8 (78%) in the rainfall runoff, started to decline after Day 50 (41%), and had a RA of 11% in both matrices by the end. In comparison, Alphaproteobacteria had the highest RA in the later stages of the experiment, with the highest abundance of all classes in the rainfall runoff on Days 105, 134, and 162. Actinobacteria reached their maximum RA (28–32%) from Day 105 onwards in the cowpats, whereas in the rainfall runoff, the maximum was reached on Day 29 (24%).

#### 3.1.3. Bacterial Taxa at the Order Level

In the cowpat, the dominant orders on Day 1 were the Oscillospirales (36%), Bacteroidales (21%), and Peptostreptococcales-Tissierellales (13%) (Figure 3, Appendix A). Overall, there were 11 orders represented at >1% RA on Day 1 in the cowpat. Eleven of these orders belonged within the Bacillota phylum: seven to the class of Clostridia (Oscillospirales, Clostridales, Peptostreptococcales-Tissierellales, Lachnospirales, Clostridia vadinBB60 group, Monoglobales, and Christensenellales), two to Bacilli (Erysipelotrichales and Acholeplasmatales), one to the Negativicutes (Erysipelotrichales), and one order to the class of Bacteroidia (Bacteroidales). By the end of the experiment on Day 162, only two of the Clostridia classes, were still ≥1% RA (Clostridiales 1% and Peptostreptococcales-Tissierellales 8%) in cowpat samples. The Oscillospirales, which were dominant on Day 1, were <1% from Day 22 onwards in the cowpat. The Bacteroidales, which were dominant on Day 1, were less than 5% RA from Day 29 onwards.

In the cowpat, other orders became more prevalent over the course of decomposition, including those belonging to the classes Actinobacteria (Micrococcales, Propionibacteriales, and Corynebacteriales), Gammaproteobacteria (Pseudomonadales and Xanthomonadales), and Alphaproteobacteria (Rhodobacterales, Rhizobiales, and Sphingomonadales), and two orders with some of the higher RAs (Flavobacteriales and Sphingobacteriales) belonging to the phylum Bacteroidota. Acholeplasmatales occurred at <8% RA in the earlier stages, especially in the cowpat, but had a very low RA thereafter. Pseudomondales were present at low RA on Day 1 and were highly variable, ranging between 4% and 20%, until starting to decline after Day 71, reaching 2–3% RA at the end of the experiment.

In cowpats, from Day 29 onwards, the following orders occurred at ≥8% RA on at least one occasion and were present at the end of the experiment at >1% RA and with the following maximum RA: Bacillales (13% on Day 50), Burkholderiales (9% on Day 71), Corynebacteriales (14% on Day 50), Flavobacteriales (17% on Day 71), Pseudomonadales (20% on Day 22), Micrococcales (10% on Day 71), Propionibacteriales (9% on Day 134), and Rhizobiales (11% on Day 134), with high abundance being noted for Sphingobacteriales (19% on Day 162). The last five orders were consistently present in the final three sampling days at RAs ranging between maximums of 7% and 19%. In addition, the Bacillales were more prevalent on Days 29 and 50 in the cowpat compared with the Clostridales before these two members of the Bacillota phylum were reduced to <5% RA for the remainder of the experiment.

In the rainfall runoff, the dominant orders on Day 1 were the Oscillospirales (34% RA), Peptostreptococcales-Tissierellales (20% RA), and Lachnospirales (8% RA), with the Bacteroidales at 17% RA, similar to the cowpat (Figure 3, Appendix A). On Day 1, there were nine orders represented at ≥1% RA in the rainfall runoff, with seven belonging to the class Clostridia, one to the Bacilli, and one to the Bacteroidia. By the end of the experiment, only the order Peptostreptococcales-Tissierellales (Clostridia) was still ≥ 1% RA. Oscillospirales, which were dominant on Day 1, were often not detected from Day 8 onwards (≤0.1%) in the rainfall runoff.

In the rainfall runoff, the initially dominant orders in the phyla Bacillota and Bacteroidota had a dramatic shift to ≤0.7% RA on Day 8 (Figure 3). By Day 8, it was the order Pseudomondales (65%) that was dominant, with minor contributions of 3–7% RA from Burkholderiales, Rhodobacteriales, Bacillales, Xanthomonadales, and Flavobacteriales. From Day 15 onwards, the RA of Pseudomonadales (22% in the rainfall runoff) decreased, and it was no longer dominant from Day 50 onwards. As the trial progressed, there was a trend for environmental bacteria, including the orders Flavobacteriales, Sphingobacteriales, Rhizobiales, and Sphingomonadales, to increase in RA to a maximum of 23% (Sphingobacteriales). On Day 1, with the exception of Flavobacteriales (0.02% RA), these bacterial orders were not detected, but they were identified on all other sampling occasions in the rainfall runoff. Bacterial taxa at the family level were not used for further analysis, such as SIMPER and FEAST; therefore, discrimination at the family level is provided in Appendix A.

#### 3.1.4. Bacterial Taxa at the Genus Level

On Day 1, the dominant phylum Bacillota (class Clostridia) was represented by two unclassified members of the order Oscillospirales (12–7% RA) and the genera *Romboutsia* (7–12%) and *Paeniclostridium* (7–8%) in cowpats and rainfall runoff. *Romboutsia* and *Paeniclostridium* (Appendix A), which belong to the order Peptostreptococcales-Tissierellales, were also prevalent in cowpat samples throughout the trial and at a lower RA in the rainfall runoff. In addition, on Day 1, the order Bacteroidales was represented by *Bacteroides,* the genus *Alistipes,* and other members of the Rikenellaceae family (range 2–3% RA) in the cowpat, and the genus *Bacteroides* (3% RA) in the rainfall runoff. In the early part of the experiment, beyond the dominant classes of Clostridia and Bacteroidia, other genera (≥1% RA) included the fecally-associated bacterium *Phascolarctobacterium* (phylum Bacillota) in the cowpat and *Turcibacter* (class Bacilli) in both matrices. In the rainfall runoff, the environmental bacterium *Stenotrophomonas* (phylum Pseudomonodota) was identified on multiple occasions except for Day 1 samples [30].

For this summertime experiment, the hottest time period occurred during sampling Days 71–134. In the cowpat, on Day 71, members of the Flavobacteriales order, *Chryseobacterium* (13%) and *Flavobacterium* (9%) (phylum Bacteroidota), were identified in the highest RA, followed by members of the Pseudomonodota (*Pseudomonas* (5%) and *Massilia* (5%). In rainfall runoff, members of the phylum Pseudomonodota, *Massilia* (maximum 24% on Day 50) and *Paracoccus* (maximum 13% on Day 50) were identified at higher RA through the middle of the experiment. The ubiquitous environmental bacteria *Pedobacter* (order Sphingobacteriales, phylum Bacteroidota) was not identified on Day 1 in either matrix but in Day 8 rainfall runoff and in both matrices from Day 15 onwards. The RA of the genus *Pedobacter* was above 5% from Day 71 onward, with >13% RA from Day 134 in the rainfall runoff.

### 3.2. Diversity

Analysis of individual replicates on each sampling day revealed low variability in diversity within the triplicate samples per treatment per sampling day (Figure 4). Overall, richness, measured by the Shannon alpha diversity index, was >4.2 for the cowpat and rainfall runoff matrices, with the highest diversity associated with fresh cowpats (Figure 4, Table 1). The exception was the rainfall runoff on Day 8, which had the lowest diversity. This shift in diversity coincided with the change in dominance from Bacillota in the rainfall runoff on Day 1 to Pseudomonodota (in particular, the Pseudomonadales) on Day 8 (Figure 1 and Figure 3).

SIMPER analysis was applied to differentiate the major taxon groups contributing to the changes in the bacterial community over time as the cowpats decomposed (Appendix A). This output was useful to identify the significant changes between days at the order level to determine which taxa were the most consequential for FEAST results (Section 3.3). The most abundant statistically significant orders (a maximum of five) between each sampling day were selected (Appendix A). Overall, 38 statistically significant orders were identified, with 23 genera with an average RA greater than 1%. Important changes in community diversity were estimated to have occurred at the phylum level between Days 1 and 8; Days 1 and 50; and Days 1 and 162 with statistically significant *p*-values (<0.05) for those orders that contributed to changing diversity at the specified time-period.

Each phylum had at least one class that contributed significantly to changing diversity at different time points within the cowpat and the rainfall matrices, except the Bacteroidota. Major significant differences between Days 1 and 8 were noted for the rainfall runoff but not the cowpat samples, suggesting that diversity changes were occurring more gradually in the cowpats as they decomposed (Appendix A). SIMPER analysis showed that these differences between Days 1 and 8 in the rainfall runoff were dominated by the orders of the Pseudomonadales (average dissimilarity of 32%) and Peptostreptococcales-Tissierellales (30%) (*p* = 0.001) (Figure 3 and Appendix A). There were also minor contributions to the average dissimilarity from the Oscillospirales and Bacteroidales (4–6%, *p* = 0.001) in the rainfall runoff. In the cowpats, these two orders represented the highest dissimilarity at 6% and 5%, respectively (*p* < 0.012). The SIMPER analysis, therefore, supported the observed minimal differences in the bacterial community between Days 1 and 8 in the cowpat runoff (Figure 1, Figure 2 and Figure 3).

In the cowpats, between Days 1 and 50, the average contribution to dissimilarity was much smaller, with Corynebacteriales (7%), Bacillales (7%), Oscillospirales (6%), Bacteroidales (4%), and Burkholderiales (4%) having the greatest statistically significant (*p* < 0.05) contributions. In contrast, in the rainfall runoff differences between Day 1 and 50, the highest statistically significant (*p* = 0.001) contributors were Peptostreptococcales-Tissierellales, Bukholderiales, Oscillospirales, Bacteroidales, and Clostridiales, contributing on average 34%, 12%, 7%, 5%, and 3% dissimilarity, respectively. In both sample matrices, Ocillospirales, Bacteroidales, and Burkholderiales were major contributors to dissimilarity; however, in cowpats, Burkholderiales contributed on average 3.4-fold less dissimilarity compared with rainfall runoff.

In both matrices, the class Clostridia was the biggest contributor to the differences between Day 1 (fresh cow feces) and Day 162 (the last day of the experiment). At the order level, Peptostreptococcales-Tissierellales contributed 22% of the average dissimilarity between the two days in the cowpats and 28% in the rainfall runoff (*p* < 0.05). The next major significant contributors in cowpats were Sphingobacterales (10%), Oscillospirales (8%), Bacteroidales (7%), and Micrococcales (4%) (*p* < 0.05). In the rainfall runoff, average dissimilarity at the order level decreased from Sphingomonadales (8%), Rhizobiales (7%), and Oscillospirales (6%) to Bacteroidales (5%) (*p* ≤ 0.005).

### 3.3. FEAST Analysis

The sources of the microbes identified by 16S rRNA amplicon sequencing in the sinks of cowpats and rainfall runoff were analyzed using the FEAST algorithm. Fecal sources from cattle collected nationwide and the soil and grass collected from within the enclosure where the cowpat experiment was located were compared. The FEAST analysis of the cowpats and rainfall runoff at the order level can be viewed in Figure 5.

It is important to choose the right taxonomic level for the interpretation of the fecal attributions for FEAST. At the lower taxon level of the genus, a lot of information is lost, and it is more difficult to discriminate fecal sources in the sinks of interest. If the taxon level is too high, the number of unknowns and errors in the taxonomy predictions is higher, increasing the noise for FEAST and making it harder to classify sources within the sinks. The smallest unknown microbial component can be observed in the order taxon of both matrices, with the rainfall runoff matrix having the lowest contribution from unknown bacteria for any matrix or taxon level (Figure 5). Other taxon levels (class and family) with higher contributions from the unknown component can be viewed in Appendix A.

On Day 1, in both matrices, the attribution to cattle feces is close to 100%, with a small proportion of unknown bacteria (Figure 5). In the rainfall runoff, the contribution from cattle fecal bacteria (order level) decreases markedly from Day 8 onwards, with attribution to the bacterial community in grass increasing in the following time periods. The rainfall runoff had a consistent grass bacterial community contribution of >75% from Day 8 to Day 134. Soil bacteria gradually increased their contribution to the bacterial community in the rainfall runoff until Day 162, when there is a marked increase to >40% attribution at both the order and class taxon levels.

In cowpats, in comparison to the rainfall runoff, there is a more gradual decline in attribution to cattle feces as the experiment progresses, showing >50% attribution till at least Day 15 (Figure 5). From Days 22 and 71 onwards, the grass was contributing over 50% of the bacterial community, with minor contributions from soil bacteria.

## 4. Discussion

Assessment of the bacterial community within cowpats is an important step to understanding the health risk associated with cowpat runoff during flood and rainfall mobilization events. Bacterial community analysis of cow feces confirmed previous studies where the majority of taxa in fresh feces on Day 1 belonged to the Bacillota (synonym Firmicutes [28]) and Bacteroidota (synonym Bacteroidetes) phyla [4,7,31,32]. The current study extended the previous metagenomic explorations of in-situ microbial communities in cowpats by sampling for almost six months post-defecation. Furthermore, the incorporation of a rainfall runoff component (20 mm/hour event) [18] assessed the mobilization of the cowpat microbiome as a contributor to fecal pollution from overland runoff into waterways. Combined, the two dominant phyla made up >90% of the phyla, while minor contributors on Day 1 included the phyla Pseudomonodota (synonym Proteobacteria) and Actinomycetota. Dominance in the community mobilized from cowpats shifted to Pseudomonodota, Actinomycetota, and environmental members of the Bacteroidota in the latter stages of decomposition for both cowpat and rainfall runoff matrices.

Differences in microbial community taxa between studies can be attributed to many factors, including sequencing methodology and farm practices. These variations in microbial taxonomy have been studied in fresh feces and the cow rumen to compare the influence of diet, farm management practices, and housing and include pasture-fed and inter-farm variability [4,5,32,33,34]. These studies have noted differing attributions to each variable; for example, a study analyzing the fecal microbiomes of cattle noted individual microbial community signatures for each farm and that the type of housing on a farm also affected this community [4]. In addition, there were overlaps between farms at the family and genus taxon levels, suggesting that these conserved taxa may have potential for fecal source tracking applications.

Identification of fecal contamination is moving beyond reliance on single genetic markers that target a group of fecal bacteria unique to an animal species, as exemplified by microbial source tracking (MST) [35,36,37]. Fecal source attribution methods for water samples are progressing to microbial community-based approaches [11,14,38,39,40]. The machine learning algorithm FEAST was employed to investigate changes in source attribution as the freshly deposited cow feces aged [13].

FEAST analysis revealed marked differences in fecal attribution between the cowpat and rainfall runoff matrices in the early stages of the experiment (Figure 5). In the fresh cowpat, bacteria were initially derived from the fecal microbes in the cow rumen. Over time, the anaerobic bacteria of the rumen were outcompeted in the cowpat by transient environmental aerobic bacteria ingested by the cow [7]. During the height of summer (Days 71–134), temperatures within the cowpat reached maximum temperatures of 45–52 °C, and the initial 90% moisture content of the cowpat was reduced to 24% by the end of the experiment [18]. The aerobic microbes present in low abundance in fresh cow feces began to increase in abundance relative to the original fecal bacteria when the community encountered stresses from those higher temperatures and low water availability (Figure 1, Figure 2 and Figure 3).

In contrast, in the rainfall runoff, FEAST analysis showed that grass/soil contributed the majority of the bacteria from Day 8 onwards. The shift to an environmental bacterial community in rainfall runoff from Day 8 suggests that after a crust formed on the cowpat, the rainfall mobilized surface and environmental microbes rather than breaking down the cowpat as occurred on Day 1. The contribution from grass and soil to the bacterial community identified in the rainfall runoff may explain the marked transition from the fecal bacterial community in the fresh cowpat to the increasing abundance of Pseudomonodota (Figure 1) such as the Rhizobiales and Sphingomonadales (Figure 3) in the latter days of the experiment. The Pseudomonadales were another order in the phylum Pseudomonodota that showed a significant increase in RA after Day 1 in rainfall runoff (65% RA). On Day 29, the Pseudomondales were observed in low abundance, suggesting that the impacts of higher temperatures and low water availability within/outside the cowpat may have contributed to a decrease in their persistence.

The addition of the soil and grass bacterial communities to the FEAST analysis highlighted the necessity for environmental sources of microflora to be added to source libraries to reduce the unknown component generated by the FEAST algorithm. In aquatic environments, reduction of the unknown source component would require understanding the types of background microflora present in waterways and including them as a “source library” for aquatic microflora. Some of these background microflorae would be derived from overland runoff and hence soil and plant microbiota, as identified in this study. Brown et al. [14], using the machine learning algorithm SourceTracker, advised using only local community fecal libraries relevant to the known fecal sources in an area. It is notable that in this current study, FEAST was run on cattle fecal samples collected nationwide, and the low variability identified for dairy cattle feces may be a consequence of the pasture-fed (rye, grass, and clover) and free-range conditions of most dairy cows in New Zealand.

FEAST analysis also noted differences at the same taxon level between the two matrices and between taxon levels within the same matrix. FEAST comparisons of class, order, and family taxon levels (Figure 5; Appendix A) showed that the ‘unknown’ component was lowest in the order taxon (Figure 5), confirming the importance of concurrently investigating multiple taxon levels when using fecal source attribution machine learning algorithms.

SIMPER analysis was applied to differentiate the major taxon groups contributing to the changes in the bacterial community over time as the cowpats decomposed. In fresh cowpats, the fecal-associated class of Clostridia comprised over 65% of the relative abundance and was the biggest contributor to the differences between Day 1 (fresh cow feces) and Day 162 (the last day of the experiment). At the order level, Peptostreptococcales-Tisserales contributed the greatest average dissimilarity between the beginning and late decomposition periods in the cowpat and the rainfall runoff. However, Oscillospirales had the highest abundance on Day 1, but were detected at <1% in the cowpat and the rainfall runoff after Days 15 and 1, respectively. This low detection rate suggests that, if identified in a waterway, Oscillospirales would be a useful indicator of fresh fecal runoff or direct fecal deposition. Further research is required to validate this assumption and determine if any members of the Oscillospirales order are specific to dairy cow feces [32]. From Day 8 onward in the cowpat, Peptostreptococcales-Tisserales was the dominant representative order of the Clostridia and was identified at 8–18% RA by the end of the experiment. Therefore, the Peptostreptococcales-Tisserales order may persist in cowpats and could be used as an indicator of aged fecal contamination in waterways, but only when detected in the absence of Oscillospirales.

From Day 8 onwards, the classes Clostridia and Gammaproteobacteria appeared to be the main contributors to differentiating between cowpats and rainfall runoff. In cowpats, the RA of Clostridia was reduced at Day 22, whereas in rainfall runoff, Clostridia’s RA decreased markedly at Day 8. Pseudonomonadales of the Gammaproteobacteria class were the most abundant order in rainfall runoff on Day 8 (>75%); however, they were not dominant in cowpats (<12%). This factor may indicate that Pseudomonadales dominate cowpat surfaces or the environment surrounding the cowpat (grass, soil). The latter is supported by the FEAST analysis on Day 8 (Figure 5) and the increasing abundance of environmentally-associated taxa with subsequent days (e.g., Rhizobiales, Rhodobacterales, and Burkholderiales) (Figure 3).

The Bacteroidia class (Bacteroidota phylum) was identified consistently in both matrices throughout the timeline of decomposition (Figure 2). A further breakdown of this class revealed that there were marked changes over time, ranging from the predominance of the fecal-associated Bacteroidales order in fresh cowpats and rainfall runoff to the increasing abundance of other orders within this class, in particular the Sphingobacteriales and the Flavobacteriales, which are linked with plant and soil environments [41,42]. This same progression was seen in the pyrosequencing 16S rRNA metagenomic study of the bacterial community mobilized from irrigated flood runoff from 2 kg cowpats decomposing over almost six months [43]. In addition, SIMPER analysis in the two matrices in the current study showed significant decreases in mobilization of the Bacteroidales order between Days 1 and 8 (Figure 3, Appendix A). Members of the Bacteroidales order are thought to be the target of the microbial source tracking (MST) markers used to discriminate bovine fecal pollution using the BacR ruminant and CowM2 quantitative PCR markers [44,45]. The study of Devane et al. [18] noted decreases in the mobilization of these two bovine MST markers from cowpats and rainfall runoff in decomposing cowpats. The CowM2 marker was the least persistent marker compared with the culturable *E. coli* and the ruminant BacR marker. CowM2 was not detected after Day 22 and Day 50 in the mobilized phases from the rainfall runoff and the cowpats (respectively). In contrast, BacR, the ruminant marker, was not identified in the cowpats until the final day of the experiment (Day 162), albeit at low concentrations of approximately 100 gene copies/100 mL. In the rainfall runoff, similarly low concentrations of BacR were identified from Days 50 to Day 134, with no detection on the last day of sampling. The decline of the mobilized portion of the Bacteroidales order, as evidenced in the cowpats and rainfall runoff (Figure 3), lends support to decreases in the detection of the bovine MST markers that target the Bacteroidales order. In particular, the low concentrations of Bacteroidales mobilized by rainfall after Day 50 may not be detected in waterways due to die-off as they travel overland in the runoff [46]. In contrast to the non-detection of the MST marker for CowM2 in aged sources, FEAST demonstrated the value of a holistic microbial community approach for identifying fecal source attributions as cattle-specific pollution was identified throughout the experiment (Figure 5). Identification of aging cattle inputs by MST methods, therefore, would need to rely on detection of the ruminant markers (e.g., BacR [44]), which only identify non-specific ruminant fecal contamination (cattle, sheep, goat, and deer).

A 57-day study [7] of bacterial community shifts in cowpats in shaded versus unshaded field conditions also noted marked decreases in the abundance of the dominant classes of Bacteroidia and Clostridia in all treatments from Day 0 to Day 2 of sampling, similar to our findings in rainfall runoff. This decrease in RA continued to Days 15–22 in both shading treatments, after which the Bacteroidia RA was minimal through the remaining 57 days. In the unshaded cowpats, which are similar to the cowpat matrix in this study, the authors [7] also noted increases in the Flavobacteria and Sphingobacteria classes belonging to the Bacteriodota phylum from Day 2 onwards. This corresponded to increases in the environmentally sourced bacteria belonging to these taxa, as observed in the current study (Figure 2 and Figure 3). Another difference between this study and Wong et al. [7] is the greater abundance of Bacilli in the unshaded cowpats after Day 0, where the Bacilli, which have capacity for aerobic respiration, became the dominant class in the Bacillota phylum from Day 2 onwards. In comparison, in the current study, the Bacilli were, in general, less abundant than the anaerobic Clostridia in the cowpat before Day 29, although the Bacillales order was more prevalent in the cowpat on Days 29 and 50. Both Clostridia and Bacilli were in low RA in the rainfall runoff from Day 8 onwards. Differences between the studies included the protection of the unshaded cowpats from rain events, whereas in this study, cowpats were exposed to all natural weather patterns. Another difference was that subsamples from individual cowpats were removed at each sampling interval so that the same cowpat was analyzed throughout the 57 days [7]. In the current study, the integrity of the cowpat was maintained as entire cowpats were analyzed at each timepoint. Maintaining whole cowpats was a strength of this study as it reduced impacts on microbial populations such as variations in temperature and changing anaerobic to aerobic conditions within the cowpat.

The differences between bacterial community studies highlight both the dominance of Bacillota and Bacteroidota phyla in the core fecal microbiota of fresh cow feces and the variability in community shifts post-defecation. This current study has revealed bacterial community shifts within the decomposing cowpats that increase our understanding of what bacteria will be available for mobilization in overland runoff/subsurface drainage resulting from floods, irrigation, and rainfall events. It has been shown that, if detected in waterways, the Oscillospirales and Bacteroidales are potential indicators of recent fecal deposition from dairy cows. In comparison, the Peptostreptococcales-Tissierellales maintained prevalence in the cowpat throughout the study and may be a potential indicator of aged fecal pollution if detected in the absence of Oscillospirales in a waterway post-flood events. Aged sources of dairy cow fecal pollution will also contain a higher proportion of environmentally-associated bacteria and potentially Bacillales if trampling/disturbance of cowpats introduces aerobic patches into cowpats. The natural microflora of the waterways is derived from multiple sources, including overland flow (plant and soil microbiota), benthic microbiota, and transient communities in the water column [8]. Additional research on the background microflora of agricultural waterways will be informed by the findings of this study and lead to improved assessments of source libraries for fecal source tracking algorithms such as FEAST.

## Figures and Tables

**Figure 1 microorganisms-11-01161-f001:**
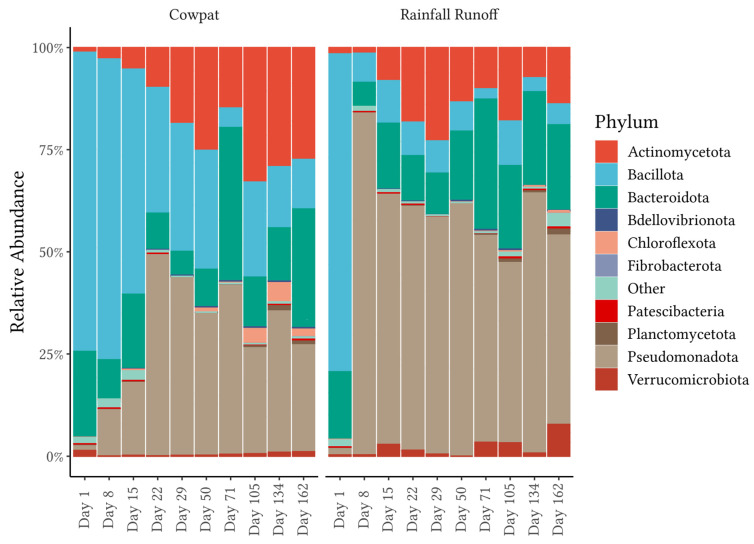
Relative abundance at the phylum level. Cowpat and rainfall runoff samples over time. Bars represent averaged replicates (*n* = 3) at each timepoint. Taxa not in the top 10 average abundances are shown as ‘Other’.

**Figure 2 microorganisms-11-01161-f002:**
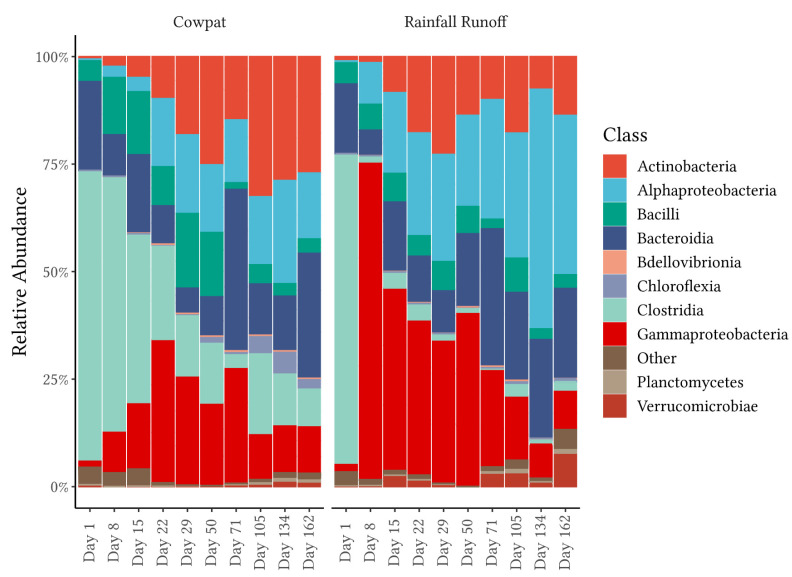
Relative abundance at the class level. Cowpat and rainfall runoff samples over time. Bars represent averaged replicates (*n* = 3) at each timepoint. Taxa not in the top 10 average abundances are shown as ‘Other’.

**Figure 3 microorganisms-11-01161-f003:**
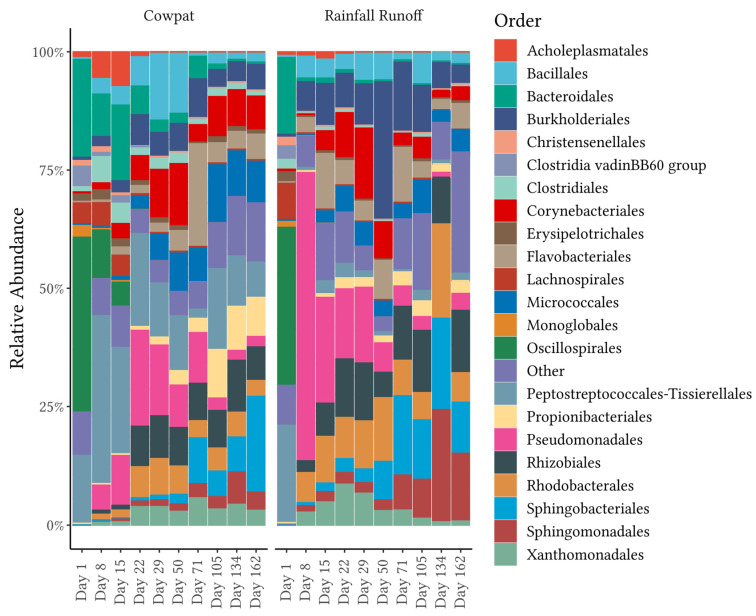
Relative abundance at the order level. Cowpat and rainfall runoff samples over time. Bars represent averaged replicates (*n* = 3) at each timepoint, and taxa with average abundances < 1% are shown as ‘Other’.

**Figure 4 microorganisms-11-01161-f004:**
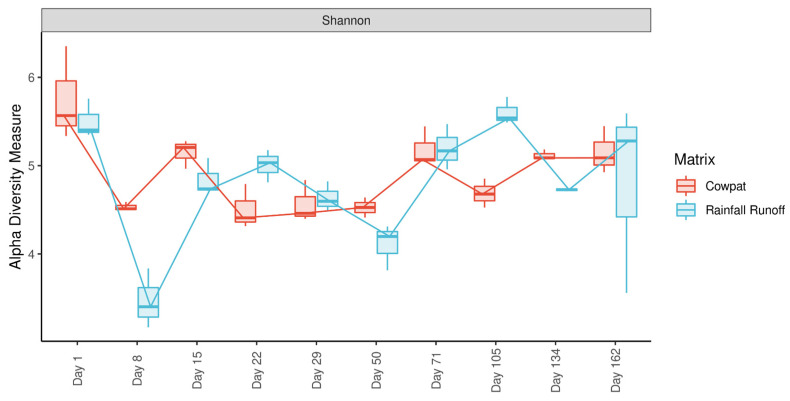
Shannon alpha diversity of matrix types over time. Lines indicate median values at each timepoint between sample types.

**Figure 5 microorganisms-11-01161-f005:**
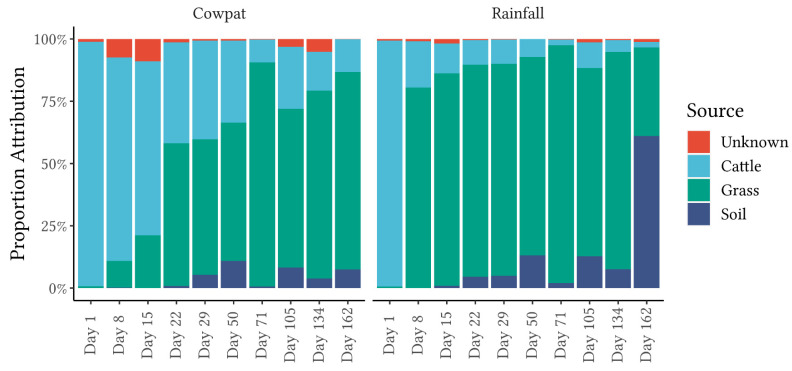
FEAST output utilizing an order-level taxonomy for cowpat and rainfall runoff samples across time. Bars represent averaged replicates (*n* = 3) at each timepoint.

**Table 1 microorganisms-11-01161-t001:** Student’s *t*-test of Shannon alpha diversity between matrix types for each day.

Day	1	8	15	22	29	50	71	105	134	162
**Cowpat Mean**	5.752	4.532	5.149	4.505	4.565	4.526	5.185	4.685	5.115	5.154
**Rainfall Mean**	5.504	3.469	4.845	5.007	4.634	4.107	5.199	5.602	4.728	4.810
** *p* ** **-value**	0.523	0.022 *	0.151	0.051	0.697	0.083	0.819	0.004 *	0.003 *	0.158

* Indicate statistically significant differences. Values are set to three significant figures.

## Data Availability

All sequence data for this project were deposited with the SRA under BioProject ID PRJNA941942.

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
