# Peer review of "Exploring the Bacterial Community in Aged Fecal Sources from Dairy Cows: Impacts on Fecal Source Tracking"

_microorganisms, 2023, doi:10.3390/microorganisms11051161_

Round 1

Reviewer 1 Report

Dear authors!

Your manuscript is interesting, describes the research strategy and results in detail, and offers a comprehensive discussion.

The impression is spoiled by the alternation of old and new Phyla names. Since the new names were published already in October 2021, I consider it necessary to change all the names of Phyla to new ones, only mention the former names in the introduction

Author Response

Point 1: The impression is spoiled by the alternation of old and new Phyla names. Since the new names were published already in October 2021, I consider it necessary to change all the names of Phyla to new ones, only mention the former names in the introduction

Response 1: SILVA database uses a mix of the old and new phyla names in their taxonomies, for example, the old Proteobacteria phylum name instead of the new Pseudomonodota but the new Bacteroidota phylum name for Bacteroidetes. Therefore, Figure 1 will be reproduced with only new phyla names and the text changed accordingly to the new phyla name where the old name will be mentioned on the first occasion it is used in the Results and in the Discussion.

Furthermore, the methods were amended at Line 197 to:

“It is noted that taxonomic name changes at the phylum level are occurring, and new phyla names according to Oren and Garrity [28] are used, with synonyms to these new names provided when first encountered in the Results and Discussion sections.”

Reviewer 2 Report

The authors used cowpats samples from previous study to analyze microbial changes during deposition. The experimental design was concise and smart, and some interesting conclusions were obtained. The paper is well structured and well written. My biggest concern is that there are only three replicates per treatment group, which is too few for microbiological analysis. The experimental methods sections in the abstract and main text are very sketchy, and the experiments are difficult to repeat by others. For example, how many cow was used for the feces collection and whether it is representative. How the rainfall runoff simulation is performed during the deposition of the cowpats, and were water and soil samples collected at a single site?

Author Response

Point 1: My biggest concern is that there are only three replicates per treatment group, which is too few for microbiological analysis. 

Response 1: 

For each treatment group (being cowpat and rainfall runoff) there were 3 samples (true replicates) per day of sampling and these were separate entire cowpats rather than three subsamples of a single cowpat.  For this time series experiment, therefore, there were 10 sampling intervals per treatment which equates to 30 samples per treatment group to allow comparisons within and between treatment groups. In addition, analysis of individual replicates on each sampling day revealed low variability in diversity within the triplicates (Figure 4).

To ensure greater clarity we improved Lines 87-92: “On each of the sampling days 1, 8, 15, 22, 29, 50, 71, 105, 134, and 162 (post-deposition), three individual, entire cowpats were sampled for analysis of the cowpat microbial reservoir, three cowpats for rainfall runoff simulation and a seventh cowpat was used to determine dry weights for moisture loss analysis. For this time series experiment, therefore, there were three cowpat samples and 10 sampling intervals per treatment (cowpat and rainfall runoff), which equates to 30 samples per treatment group.”

And added Line 356: “Analysis of individual replicates on each sampling day revealed low variability in diversity within the triplicate samples per treatment per sampling day (Figure 4).”

Point 2: 

The experimental methods sections in the abstract and main text are very sketchy, and the experiments are difficult to repeat by others. For example, how many cow was used for the feces collection and whether it is representative. How the rainfall runoff simulation is performed during the deposition of the cowpats, and were water and soil samples collected at a single site?

Point 2a: Revise the methods section in the abstract

Response 2a:

“2) Methods: The study examines changes in the bacterial community available for mobilization from in-situ decomposing cowpats, and the effects of simulated rainfall. The microbiome of individual cowpats was monitored over five-and-a-half-months. We used 16S rRNA metagenomics and the machine learning software, FEAST (Fast Expectation-mAximization for microbial Source Tracking) for bacterial and fecal source assignments.”

Point 2b: how many cow was used for the feces collection?

Response 2b: We have added in the number of cows to Line 80 in methods

Line 80:  “ In brief, fresh fecal material from 60 pasture-fed dairy cows was collected from the concrete pad leading into the milking shed which had been washed down prior to milking.”

Point 2c: How the rainfall runoff simulation is performed during the deposition of the cowpats,

Response 2c: This section has been rewritten (Line 115) to improve understanding of the methodology. For further details, the reader is also referred to the original paper (reference 18, Devane et al. 2022).

Line 115: “A rainfall simulator was constructed [18] which delivered a rainfall event of 20 mm/hr with formation of <2 mm raindrops at terminal velocity [19]. In brief, sterile distilled water (1146 mL) (MilliQ, Merck Millipore, Darmstadt, Germany) was added to a circular drip tray containing 25 needles of 20-gauge size (nominal internal diameter of 0.603 mm). On each sampling occasion, three cowpats were individually collected by lifting up the nybolt mesh plus cowpat and transferring to a pre-weighed 450 x 300 mm length tray. This tray with the cowpat had an approximate 10% slope to facilitate collection with four 10 mm holes and nine 3 mm holes drilled at one end to allow the rainfall runoff to flow through a sterile funnel into a sterile 500 mL polypropylene collection bottle. The bottles were placed into holes in the ground to directly capture the rainfall runoff. The drip tray with needles was placed 92 cm above the cowpat samples and wind disturbance was minimized by wrapping the rainfall simulator in plastic [18]. The measured volume of collected runoff was recorded and DNA extraction performed as outlined in the following section.”

Point 2d: were water and soil samples collected at a single site?  

Response 2d: We apologize, but the authors were unsure of what Reviewer 2 was referring to as water samples were not collected in this study, only samples from cowpats resuspended in water and simulated rainfall runoff from cowpats.

Line 94 was amended to try to make clearer the collection of environmental samples for the FEAST source library:

“Ten samples of grass (ryegrass and clover) and underlying soil (top 2 cm) were collected for bacterial community analysis for the FEAST source library. These environmental samples were collected from the same secure enclosure where the cowpat experiment was undertaken.”

Reviewer 3 Report

 The manuscript of Devane et al. entitled ‘Exploring the Bacterial Community in Aged Fecal Sources from Dairy Cows: Impacts on Fecal Source Tracking’ in my opinion is a comprehensive manuscript and the aim is significant especially due to the scale of dairy cow fecal pollution. The study explores the fecal microbiome of cattle and the potential ecological implications of aging fecal pollution on waterways and despite that the scope may seem trivial, due to the use of advanced high-throughput analysis techniques and modeling, this work sheds new light on the complexity of the studied topic.

In my opinion, the manuscript is prepared accurately and legibly. Some minor punctuation errors do not affect the quality of the reception. Figures are prepared legibly and carefully. The methods are described transparently and reproducibly. The conclusions were supported by the results and discussed properly with adequate and relevant references.

The design of the research does not raise any doubts and the number of repetitions is also sufficient.

I have only two minor concerns about

1)            R v4.0.5 and the R package vegan v2.6-2 were applied for tests. I see the provided reference but is that the best option for the animal origin data analysis?

2)            Is the microflora word, applied for the first time in the 407 line, used knowingly and intentionally?

I would recommend accepting the manuscript after minor revision

Author Response

Point 1: R v4.0.5 and the R package vegan v2.6-2 were applied for tests. I see the provided reference but is that the best option for the animal origin data analysis?

Response 1: The vegan package was used to apply SIMPER analysis to determine which taxa contributed most to the dissimilarity between time points as the samples aged over time. The authors have re-written the text to clarify the process and show that the vegan package was not used for determining the origin of the taxa but was only used to investigate the key players involved in community dissimilarity during the experiment.

Line 203 Changed to:

“2.8 Statistical Analyses

Statistical analyses were carried out in R v4.0.5 and using the R package vegan v2.6-2 [28]. A Shapiro-Wilks test was performed to test for normality, and a Student’s t-test was used to test for significance between the Shannon diversity of samples with the R stats package. SIMPER analysis using vegan v2.6-2 identified the taxa contributing to dissimilarities between sampling days to test the significance of differences between sampling intervals over time [28].”

Point 2: Is the microflora word, applied for the first time in the 407 line, used knowingly and intentionally?

Response 2: Now starting at line 452 in Results: microflora has been changed to "bacteria" or "bacterial community" when referring directly to results from this study.

And then from line 507 in Discussion the authors have changed microflora to bacteria or bacterial community when referring directly to bacteria only. The exception is for the paragraph starting at Line 530 and also lines 646 to 650, where the authors have intentionally used microflora to refer to all microbes (bacteria, protozoa, fungi etc.).